# Recent Progress in Treatment for HER2-Positive Advanced Gastric Cancer

**DOI:** 10.3390/cancers16091747

**Published:** 2024-04-30

**Authors:** Takeshi Kawakami, Kentaro Yamazaki

**Affiliations:** Division of Gastrointestinal Oncology, Shizuoka Cancer Center, Shizuoka 411-0934, Japan; k.yamazaki@scchr.jp

**Keywords:** ERBB2, gastric cancer, HER2 overexpression, immunotherapy, treatment advance

## Abstract

**Simple Summary:**

Human epidermal receptor (HER) 2-positive advanced gastric cancer (AGC) is one of the major subtypes of gastric cancer, accounting for ~20% of all cases. After the success of a ToGA trial, the development of treatment for HER2-positive AGC had been at a standstill for a long time. Recently, trastuzumab deruxtecan in later-line therapy and the combination therapy of chemotherapy plus trastuzumab with immune checkpoint inhibitor has demonstrated significant survival benefit in first-line treatment. Currently, several clinical trials of new types of anti-HER2 agents are ongoing.

**Abstract:**

Human epidermal receptor (HER) 2-positive advanced gastric cancer is one of the major subtypes of gastric cancer, accounting for ~20% of all cases. Although combination therapy with trastuzumab and chemotherapy provides meaningful survival benefit, clinical trials targeting HER2 have failed to demonstrate clinical benefits in first- or subsequent-line treatment. Trastuzumab deruxtecan, an antibody–drug conjugate, has shown positive results even in later-line treatment and has become new standard treatment. In first-line therapy, combination therapy with pembrolizumab and trastuzumab plus chemotherapy demonstrated a dramatic response rate. Therefore, the FDA rapidly approved it without waiting for the results of survival time. The emergence of combination therapy including immunotherapy with HER2-targeting agents and the development of HER2 targeting agents with or without immunotherapy have been advancing for treating HER2-positive gastric cancer. In this review, we will discuss the current status of treatment development and future perspectives for HER2-positive gastric cancer.

## 1. Introduction

Advanced gastric cancer (AGC) with overexpression of human epidermal growth factor receptor 2 (HER2), which is currently one of the major subtypes of AGC, was first reported in 1986 [1]. HER2 is a transmembrane tyrosine kinase receptor and a member of the epidermal growth factor receptor (EGFR) family (HER1, HER2, HER3, and HER4). Ligand binding to HER induces a signaling cascade, promoting cell proliferation through the RAS/MAPK pathway and inhibiting cell death through the PI3K/AKT pathway [2]. HER2 overexpression is diagnosed with immunohistochemistry (IHC) and gene amplification in situ hybridization (ISH) according to the following guidelines: IHC 3+ or 2+/ISH+ (chromosome enumeration probe [CEP] 17 ratio ≥ 2) is defined as HER2-positive gastric cancer [3]. The prognostic role of HER2 is still controversial [4,5].

The development of AGC treatment has progressed dramatically in the last decade. Trastuzumab is a human monoclonal IgG antibody that inhibits homodimerization and heterodimerization with HER2, cleaves the extracellular matrix, and induces antibody-dependent cellular cytotoxicity (ADCC). The antitumor activity of combination therapy with trastuzumab and chemotherapy for HER2-positive AGC was first demonstrated in the ToGA trial [6]. Combination therapy with trastuzumab and chemotherapy showed significantly longer overall survival (OS) compared with chemotherapy alone. Based on the results of the ToGA trial, trastuzumab is used as standard first-line treatment for HER2-positive AGC worldwide. Although several clinical trials of anti-HER2 therapy, which has shown survival benefits for breast cancer, were conducted in a first-line setting after the ToGA trial, none showed survival benefits for HER2-positive AGC (Table 1). Recently, pembrolizumab, a programmed cell death-1 (PD-1) antibody, plus trastuzumab and chemotherapy, showed drastic high overall response rate (ORR) and significantly improved progression-free survival (PFS) in the KEYNOTE-811 trial [7]. Several clinical trials combining immunotherapy with HER2-targeting agents are in progress.

Developing treatment for patients with HER2-positive AGC previously treated with trastuzumab is an important challenge. Several clinical trials involving patients previously treated with a trastuzumab-containing regimen failed to demonstrate survival benefits (Table 2). Spatial and temporal heterogeneity are considered one of the resistant mechanisms [14]. In gastric cancer, the distribution patten of HER2-positive cancer cells and/or concomitant genetic alterations vary in tumor tissues between primary and metastatic lesions. Furthermore, HER2-positive tumor cells are either eliminated in a first-line trastuzumab-containing regimen (HER2 loss) and/or acquire genomic alterations. HER2 loss was observed in 30–70% of the patients who progressed after a trastuzumab-containing regimen [12,15,16]. The development of drugs that overcome resistance is challenging. Recently, trastuzumab deruxtecan (T-DXd), an antibody drug conjugate (ADC), has significantly improved ORR and prolonged survival compared with conventional chemotherapy (irinotecan or paclitaxel) in the DESTINY-Gastric01 study [13]. Many types of anti-HER2 agents are being developed, and the development of therapeutics for HER2-positive AGCs is in progress (Figure 1).

## 2. Trastuzumab

### 2.1. First-Line

Trastuzumab is a humanized monoclonal IgG antibody that binds HER2 receptors, inhibiting receptor internalization and promoting the transduction of cell proliferation signaling. In addition, trastuzumab has ADCC activity. Chemotherapy plus trastuzumab significantly improved OS in all randomized patients as primary endpoint compared with chemotherapy alone (13.8 months vs. 11.1 months; hazard ratio [HR] 0.74; 95% confidence interval [CI] 0.60–0.91) in the ToGA trial [6]. Preplanned exploratory analysis of patients in the HER2-positive cohort revealed better antitumor effects from trastuzumab combination therapy in OS (16.0 months vs. 11.8 months; HR: 0.65; 95% CI: 0.51–0.83). Combination chemotherapy with trastuzumab also contributed to quality of life. Quality of life was assessed in the ToGA trial as one of the secondary endpoints using the European Organization for Research and Treatment Cancer (EORTC) quality of life questionnaires QLQ-C30 and QLQ-STO22. QLQ-C30 and QLQ-STO22 scores showed a favorable trend in trastuzumab plus chemotherapy compared to chemotherapy alone. In addition, the time to a 10% definitive deterioration in EORTC QLQ-C30 global health status in chemotherapy plus trastuzumab was significantly longer than that in chemotherapy alone (12.1 months vs. 6.8 months; *p* < 0.0001) [17].

Trastuzumab is the first molecular targeting agents showing the anti-tumor effects of AGC; however, several patients were thought to have initial resistance for trastuzumab and clinical trials continued.

For untreated HER2-positive AGC, two phase 3 trials were conducted after the ToGA trial. The JACOB trial investigated the additional effect of pertuzumab, a humanized monoclonal HER2-targeting antibody, which inhibits the heterodimerization of HER2 and HER3, on chemotherapy plus trastuzumab [9]. Chemotherapy plus trastuzumab and pertuzumab treatment did not provide meaningful survival benefit in OS as the primary endpoint compared with chemotherapy plus trastuzumab (17.5 months vs. 14.2 months: HR: 0.84; 95% CI: 0.71–1.00; *p* = 0.057). However, the combination therapy of pertuzumab demonstrated significant improvement in PFS (8.5 months and 7.0 months (HR: 0.73; 95% CI: 0.62–0.86; *p* = 0.001) and ORR (56.7% vs. 48.3%; *p* = 0.026). The TRIO-013/LOGiC trial investigated capecitabine, oxaliplatin plus lapatinib (lapatinib group), a tyrosine kinase inhibitor (TKI) of EGFR and HER2, compared with chemotherapy alone. The lapatinib group did not exhibit prolonged OS as primary endpoint compared with chemotherapy alone (12.2 months vs. 10.5 months; HR: 0.91; 95% CI: 0.73–1.12), whereas PFS (6.0 months vs. 5.4 months; HR: 0.82; 95% CI: 0.68–1.00; *p* = 0.0381) and ORR (53% vs. 39%; *p* = 0.0031) of the lapatinib group showed significant improvement [8,9]. Unlike that for metastatic breast cancer, both trials of experimental treatment failed to demonstrate statistically significant improvement in OS. The possible reason for the negative results is the difference in biology between breast cancer and gastric cancer: intratumoral heterogeneity of HER2 expression and discordant HER2 expression between primary and metastatic lesions [14]. As for the JACOB trial, the anti-tumor effect of lapatinib may be lower than that of trastuzumab. In a phase 3 trial comparing trastuzumab with lapatinib with taxane-based chemotherapy for metastatic breast cancer, lapatinib showed significantly shorter survival than trastuzumab [18]. Progress in the development of treatments for HER2-positive gastric cancer came to a standstill for some time.

Recently, the development of combination therapy with trastuzumab and immune checkpoint inhibitors has been in progress. Platinum agents, such as oxaliplatin, are known to induce immunogenic cell death, which triggered an immune response, and enhance the anti-tumor effect of immune checkpoint inhibitors [19]. For HER2-negative AGC, the combination therapy of chemotherapy with nivolumab has already provided the significant improvement of survival against chemotherapy alone in the CheckMate-649 and ATTRACTION-4 trial [20,21]. For HER2-positive AGC, trastuzumab administration is reported to have a positive effect on anti-tumor immune response. In a preclinical trial, trastuzumab promoted the uptake of soluble HER2 into dendritic cells through FcR, inducing HER2-specific cytotoxic T lymphocytes and promoting cross-presentation by dendritic cells [22]. Trastuzumab induced resistance by upregulating the expression of programmed cell death-ligand 1 (PD-L1), and inhibition of the binding between PD-1 and PD-L1 is considered beneficial treatment [23]. Based on the rationale described, several clinical trials of combination therapy with pembrolizumab, a PD-1 antibody, with chemotherapy and trastuzumab were conducted in USA and countries in Asia, demonstrating higher ORR and tumor burden reduction [24,25,26]. A phase 3 trial was conducted to evaluate the superiority of chemotherapy plus trastuzumab with pembrolizumab (pembrolizumab group) against chemotherapy plus trastuzumab with placebo (placebo group) for untreated HER2-positive AGC (KEYNOTE-811) [7]. At second interim analysis, median PFS was significantly longer in the pembrolizumab group than the placebo group (10.0 months vs. 8.1 months; HR: 0.72; 95% CI: 0.60–0.87; *p* = 0.0002 [superiority boundary, 0.0013]). Although median PFS was also longer in the pembrolizumab group than the placebo group (10.8 months vs. 7.2; HR: 0.70; 95% CI: 0.50–0.85) in patients with a PD-L1 combined positive score (CPS) ≥ 1, median PFS did not differ in patients with CPS < 1 (9.5 months vs. 9.6 months; HR: 1.17; 95% CI: 0.73–1.89). At second interim analysis, median OS was not significantly longer in the pembrolizumab group than the placebo group (20.0 months vs. 16.9 months; HR: 0.87; 95% CI: 0.72–1.06; *p* = 0.084). In the pembrolizumab group, significant improvement in OS was not observed even for patients with CPS ≥ 1. Although the OS of the pembrolizumab group was not significantly improved compared with the placebo group at third interim analysis, the trial prepares for the final analysis based on the recommendation of the independent data monitoring committee. ORR, which was the secondary endpoint, was significantly higher in the pembrolizumab group than the placebo group (72.6% vs. 59.8%). Similar to PFS and OS, ORR in CPS < 1 group was comparable between the pembrolizumab and placebo groups (69.2% vs. 67.3%). The median duration of response was longer in the pembrolizumab group than the placebo group (11.2 months vs. 9.0 months). The frequency of treatment-related adverse events was similar in both group: any grade: 97% vs. 97% and grade ≥3: 58% vs. 51%. No new safety signals were observed. The FDA and EMA, as well as Brazil, have approved chemotherapy plus trastuzumab with pembrolizumab, and the approval process is progressing worldwide.

A randomized phase 2 study was performed to evaluate the efficacy of trastuzumab plus nivolumab with ipilimumab (ipilimumab group) or FOLFOX (FOLFOX group) for untreated HER2-positive AGC [27]. The primary endpoint was OS rate at 12 months, which was 57% (95% CI: 41–71%) for the ipilimumab group and 70% (95% CI: 54–81%) for the FOLFOX group. Although the OS rate of the FOLFOX group reached predefined criteria, that of ipilimumab did not. As for safety, the incidence rate of treatment related severe adverse events was 39% and 35%. Grade ≥3 immune related adverse events (hepatitis, colitis, pneumonitis, and endocrine disorders) were less than 10%. Toxicities of ipilimumab group was manageable. This was the first trial to evaluate the significance of a chemo-free regimen using immunotherapy for AGC; the survival benefit of this chemo-free regimen was not demonstrated.

### 2.2. Previously Treated with a Trastuzumab-Containing Regimen after Second-Line Treatment

Several clinical trials targeting patients previously treated with trastuzumab failed to demonstrate the survival benefit of trastuzumab beyond progression (Table 2). Although lapatinib, a small-molecule TKI of EGFR and HER2, plus paclitaxel demonstrated significantly higher ORR than paclitaxel alone, lapatinib plus paclitaxel failed to improve OS as primary endpoint in the TyTAN trial (11.0 months vs. 8.9 months; HR: 0.84; 95% CI: 0.64–1.11) [10]. Trastuzumab emtansine (T-DM1), an ADC where trastuzumab is linked to the tubulin inhibitor emtansine by a stable linker, failed to prolong OS against paclitaxel monotherapy in the GATSBY study (7.9 months vs. 8.6 months; HR: 1.15; 95% CI: 0.87–1.51; *p* = 0.86) [11]. The spatial and temporal heterogeneity of HER2 expression is thought be one of the reasons for the failure of these studies. In the GATSBY study, biomarker analysis showed that the treatment efficacy of T-DM1 was associated high HER2 expression, and differences in antitumor effects based on the presence or absence of *PTEN* and *PIK3CA* mutations, which are involved in trastuzumab resistance, were not significant. However, patients with high cMET expression (IHC 3+) tended to show better treatment effects with T-DM1 compared with paclitaxel [28]. The T-ACT study evaluated the significance of continuous use of trastuzumab beyond progression (TBP), comparing paclitaxel plus trastuzumab with paclitaxel alone in the second-line setting [12]. Unfortunately, the T-ACT study failed to demonstrate the survival benefits of TBP. In this study, HER2 expression was examined before initiating second-line treatment if tumor tissue sample was obtained. The results showed that the antitumor activity of paclitaxel plus trastuzumab was not associated with HER2 expression, although sample size was small. Interestingly, subgroup analysis showed a trend toward better outcomes with paclitaxel plus trastuzumab in patients with longer interval between final dose of trastuzumab and paclitaxel plus trastuzumab.

The lack of antitumor effect of trastuzumab is not solely due to heterogeneity; crosstalk between the HER2 pathway and angiogenesis is also a possible resistance mechanism [29]. Based on this hypothesis, an open label, multicenter, phase 1b/2 trial was conducted to explore the antitumor effect of paclitaxel and ramucirumab with trastuzumab in Korea [30]. Paclitaxel plus ramucirumab with trastuzumab was administered according to the following dosing regimen: paclitaxel 80 mg/m^2^ on days 1, 8, and 15; ramucirumab 8 mg/kg on days 1 and 15 every 4 weeks; and trastuzumab 4 mg/kg for the loading dose on cycle 1, day 1, and 2 mg/kg for the subsequent dose. Patients who experienced failure with a trastuzumab-containing regimen as first-line treatment were included. HER2-positive status was determined before first-line treatment. At a median follow-up time of 27.5 months, median PFS and OS were 7.1 and 13.6 months, respectively. ORR and disease control rate were 54% and 96%, respectively. In this study, 23 tumor samples were obtained from patients after refractory to first-line treatment. Differences in PFS and OS between HER2 status before second-line treatment were not significant. The blockade of HER2 and angiogenic pathways can overcome the resistance mechanism of trastuzumab.

Treatment focusing on another immune checkpoint is in progress to overcome trastuzumab resistance. The CD47/SIRPα pathway is considered an immune checkpoint in innate immunity as well as PD-1/PD-L1 axis in acquired immunity. Binding of overexpressed CD47 on tumor cells with SIRPα on macrophage surface stimulates the signaling for phagocytosis suppression [31]. Evorpacept––an engineered protein comprising a high-affinity CD47 blocker fused with an inactive IgG Fc region––inhibits the signal for phagocytosis suppression. Combination therapy with drugs containing active Fc domains, such as trastuzumab, is expected to enhance phagocytic activity. A phase 1b trial of evorpacept plus pembrolizumab or trastuzumab was conducted [32]. This trial included non-small cell lung carcinoma, head and neck squamous cell carcinoma, and HER2-positive gastric or esophagogastric junction cancer. HER2-positive gastric or esophagogastric junction cancer received evorpacept plus trastuzumab. In the dose-expansion cohort, ORR was 21.1% (95% CI: 6.1–45.6) and median duration of response was 8.7 months, for HER2-positive gastric or esophagogastric junction cancer. No new safety signal was observed. For second-line treatment, a phase 2/3 trial investigating the superiority of paclitaxel, ramucirumab, trastuzumab plus evorpacept compared with paclitaxel plus ramucirumab is ongoing (NCT05002127).

## 3. Trastuzumab Deruxtecan

Trastuzumab deruxtecan (T-DXd) is a HER2-targeted ADC. It combines an anti-HER2 humanized monoclonal antibody with approximately eight molecules of deruxtecan––a campthothecin derivative that inhibits topoisomerase I––linked through a tumor cell-internalizing peptide-containing linker [33]. T-DXd specifically binds to HER2 and is internalized within tumor cells, where the linker is cleaved by lysosomal enzymes to release deruxtecan, which inhibits topoisomelase I, leading to DNA damage, apoptosis, and inhibition of tumor cell proliferation. T-DXd has bystander effect––it affects HER2-negative tumor cells around HER2-positive cells. This effect is expected to overcome intratumor heterogeneity.

A randomized phase 2 trial, the DESTINY-Gastric01 study, was conducted in Japan and South Korea to compare the antitumor effect of T-DXd with physician’s choice chemotherapy (irinotecan or paclitaxel) for HER2-positive AGC previously treated with ≥2 lines of treatment, including fluoropyrimidine, a platinum agent, and trastuzumab [13]. T-DXd demonstrated significantly higher ORR as the primary endpoint than physician’s choice chemotherapy (paclitaxel or irinotecan monotherapy) for HER2-positive AGC previously treated with a trastuzumab-containing regimen in later-line treatment (51% vs. 14%; *p* < 0.001). Median OS was significantly longer in the T-DXd group than the physician’s choice chemotherapy group (12.5 months vs. 8.4 months; HR: 0.59; 95% CI: 0.39–0.88; *p* = 0.01).

Almost the entire safety profile of T-DXd was manageable; however, interstitial lung disease (ILD) required careful management. ILD was observed in 10% (12) of the patients, with no treatment-related deaths in the DESTINY-Gastric01 study. In the DESTINY-Breast01 study, a phase 2 study evaluating the efficacy of T-DXd for advanced breast cancer, any grade of ILD was observed in 13.6% (25) of the patients, including grade 5 in 2.2% (4) of the patients. Higher dose of T-DXd and Japanese patients are reported to be risk factors for ILD [34]. Early diagnosis with appropriate assessment of symptoms, laboratory tests, and imaging is very important to prevent severe ILD. Furthermore, nausea and vomiting require careful consideration. The DESTINY-Gastric01 study did not regulate the management of prophylaxis for emesis and the incidence of nausea and vomiting were 63% with grade 35% and 26% with no grade 3 or 4. National Comprehensive Cancer Network (NCCN) guidelines recommend treating T-DXd as a high emetic risk agent [35]. It recommends the prophylaxis use of a combination of dexamethasone, serotonin receptor antagonist, and/or neurokinin-1 receptor antagonist.

The DESTINY-Gastric01 study evaluated the antitumor effect of T-DXd for patients with HER2-low expression as the exploratory cohort who progressed during or after at least two prior regimens without trastuzumab. The exploratory cohort was divided into two types––cohort 1: IHC2+/ISH− and cohort 2: IHC1+ [36]. ORR was 26.3% in cohort 1 and 9.5% in cohort 2. Median PFS and OS were 4.4 and 7.8 months in cohort 1 and 2.8 and 8.5 months in cohort 2, respectively. Unlike in breast cancer, T-DXd did not yield the expected antitumor effects in the HER2-low group. Because HER2-low AGC has worse prognosis than HER2-positive [37], the HER2-low group is considered a clinical unmet need and further treatment development is deemed necessary. A randomized trial with a larger cohort must estimate the antitumor effect of T-DXd for HER2-low AGC.

A single-arm phase 2 trial––the DESTINY-Gastric02 study––targeting patients with HER2-positive AGC who progressed after a trastuzumab-containing regimen as first-line treatment was conducted in USA and Europe [38]. The confirmation of HER2-positive using biopsy sample of post-trastuzumab progression was needed in this trial. ORR as the primary endpoint of T-DXd was 38% (95% CI: 27.3–49.6): lower limit of 95% CI: exceeded 27% as predefined threshold. Median PFS and OS were 5.6 and 12.1 months, respectively. Although the DESTINY-Gastric02 study tended to have lower ORR than the DESTINY-Gastric01 study, their survival times were comparable. T-DXd demonstrated clinically meaningful antitumor effects even in the second-line setting. A randomized phase 3 study evaluating the efficacy and safety of T-DXd compared with ramucirumab plus paclitaxel for patients with HER2-positive AGC who progressed on or after a trastuzumab-containing regimen as first-line treatment, the DESTINY-Gastric04 study is ongoing (NCT04704934).

T-DXd is expected to be effective not only in second-line and subsequent treatments, but also in first-line therapy, when used in combination with immune checkpoint inhibitors and cytotoxic anticancer agents. A phase 1b/2 study is being conducted to explore the effectiveness of T-DXd in combination with cytotoxic agents and/or immunotherapy for HER2-positive AGC (DESTINY-Gastric03: NCT04379596). This study has three parts: part 1 is for HER2-positive AGC after progression on at least one prior regimen with trastuzumab; part 2 and 3a are for previously untreated HER2-positive AGC; and part 3b is for previously untreated HER2-low (ISH 2+/ISH− or IHC 1+) AGC. This study will be pivotal in determining the direction of treatment development for earlier-line therapy in HER2-positive AGC and HER2-low cases.

## 4. Zanidatamab

Zanidatamab is a humanized, bispecific, IgG1-like monoclonal antibody that binds to the juxtamembrane extracellular domain (ECD4) and dimerization domain (ECD2) of HER2. Zanidatamab binds various ranges of HER2 expression (low to high) and downregulates HER2 on the cell surface by inducing the formation of receptor clusters and promoting receptor internalization. Zanidatamab inhibits growth factor-dependent or -independent proliferation of tumor cells and potently activates ADCC, antibody-dependent cellular phagocytosis, and complement-dependent cytotoxicity [39]. A phase 1 trial of zanidatamab for HER2-expressing or *HER2*-amplified cancers, including AGC, demonstrated clinically meaningful antitumor effects [40]. Diarrhea was observed in ~50% of the cases, and almost all of the cases were grade 1–2. A phase 2 study was conducted to evaluate the antitumor activity of zanidatamab plus standard first-line chemotherapy (capecitabine and oxaliplatin [CAPOX], 5-fluorouracil/leucovorin and oxaliplatin [mFOLFOX6], or 5-fluorouracil and cisplatin [FP]) [41]. Confirmed ORR as primary endpoint was 79%, and median duration of response was 20.4 months. PFS and OS were 12.5 months and not estimable, with a median follow-up of 26.5 months, respectively. Diarrhea was the most frequent treatment-related adverse event: any grade in 93% and grade ≥ 3 in 35% of the cases. Prophylactic use of antidiarrhea drug was required during the trial. The incidence of diarrhea decreased from 52% to 14% after an antidiarrhea drug was used for prophylaxis. A phase 3 trial comparing zanidatamab plus CAPOX, zanidatamab, and tislelizumab plus CAPOX with trastuzumab plus CAPOX is ongoing (NCT05152147).

## 5. Margetuximab

Margetuximab is an Fc-engineered HER2 monoclonal antibody. The Fc region of margetuximab was enhanced by substitutions in five amino acids, resulting in increased binding affinity to CD16A (FcγRIIIa) and decreased binding affinity to CD32B (FcγRIIb), which has inhibitory effects on ADCC, antibody-dependent cellular phagocytosis, and B cell activation [42]. Although trastuzumab has ADCC activity through NK cells, this modification showed stronger cytotoxic effects compared with trastuzumab––margetuximab promoted PD-1/PD-L1 and lymphocyte activation gene-3 (LAG-3) expression on NK and NK T cells [43,44].

A first-in-human phase 1b trial of margetuximab combination therapy was conducted for HER2-positive advanced solid tumors, including gastric cancer [45]. Margetuximab demonstrated significantly higher cytotoxicity than trastuzumab. ORR was 12% and mPFS was 14 weeks; nevertheless, the participants were heavily treated. A phase 1b/2, dose escalation study of margetuximab in combination with pembrolizumab was conducted in patients with relapsed or refractory HER2-positive gastroesophageal adenocarcinoma (CP-MGAH22-05) [46]. Despite a chemotherapy-free regimen and heavy pretreatment, margetuximab plus pembrolizumab demonstrated a median OS of 12.5 months, with no new safety signals. Although ORR was 18% for the response-evaluable population, ORR was 44% for patients with both HER2 IHC 3+ and PD-L1-positive (CPS ≥ 1). No patient with IHC 2+, regardless of CPS status, responded to martgetuximab plus pembrolizumab. Based on the results of CP-MGAH22-05, a phase 2/3 study is ongoing with a combination of margetuximab and an immune checkpoint inhibitor plus chemotherapy for treating naïve HER2-positive AGC compared with chemotherapy plus trastuzumab (MAHOGANY trial: NCT04082364) [44]. This study comprises cohorts A and B. Cohort A includes a single experimental arm: margetuximab plus retifanlimab, administered to patients with IHC 3+, CPS ≥ 1, and nonmicrosatellite instability-high (non-MSI-H). Cohort B consists of three experimental arms: margetuximab plus chemotherapy with retifanlimab, margetuximab plus chemotherapy with tebotelimab, and margetuximab plus chemotherapy, administered to patients with IHC 3+ or IHC 2+/ISH+ regardless of CPS. Retifanlimab is an anti-PD-1 monoclonal antibody, and tebotelimab is a bispecific antibody that binds both PD-1 and LAG-3. The coexpression of PD-1 and LAG-3 is observed in tumor-infiltrating lymphocytes and contributes to T-cell exhaustion; coinhibition of PD-1 and LAG-3 showed synergistic response in preclinical models [47,48]. For untreated advanced melanoma, combination therapy with relatlimab, anti-LAG-3 antibody plus nivolumab significantly improved PFS as the primary endpoint (HR: 0.75; 95% CI: 0.62–0.92; *p* = 0.006) compared with nivolumab alone [49]. Thus, combination therapy with margetuximab and dual blockade of PD-1 and LAG-3 is expected to have higher antitumor activity than treatment with margetuximab alone. In cohort A, tumor shrinkage was seen in 78% of the patients; ORR was 53% and median duration of response was 10.3 months for response-evaluable non-MSI-H [50]. Median PFS was 6.4 months and median OS was not reached. This chemotherapy-free regimen demonstrated a favorable antitumor effect with manageable toxicities; however, the sponsor of this study decided to discontinue cohort A part 2 for business reasons.

## 6. Tucatinib

Tucatinib is an orally administered small-molecule TKI that reversibly blocks HER2 tyrosine kinase activity with higher selectivity compared with other HER2-targeting TKIs, such as lapatinib and neratinib. Tucatinib plus trastuzumab demonstrated higher antitumor activity in a preclinical study, including HER2-positive gastric cancer PDX models [51]. Combination therapy with tucatinib and trastuzumab plus capecitabine significantly prolonged PFS as the primary endpoint (HR: 0.54; 95% CI: 0.42–0.71; *p* < 0.001) in patients with HER2-positive metastatic breast cancer who had a history of anti-HER2 therapy compared with trastuzumab plus capecitabine [52]. Furthermore, tucatinib plus trastuzumab showed a clinically meaningful antitumor effect in patients with HER2-overexpressing metastatic colorectal cancer who were previously untreated with anti-HER2 therapy (ORR: 38.1% ([complete response 4% and partial response 35%]), with manageable toxicities in a phase 2 trial [53]. Superior antitumor effects were demonstrated regardless of initial or prior anti-HER2 therapy. For gastric cancer, a phase 1b/2 trial of tucatinib plus trastuzumab and oxaliplatin-based chemotherapy or pembrolizumab-containing combinations for HER2-positive gastrointestinal cancers is ongoing (NCT04430738).

## 7. Conclusions and Future Directions

Although many clinical trials of novel agents targeting HER2 with or without immune checkpoint inhibitors are going on, there are several challenges to making advances for HER2-positive AGC. Combination therapy with chemotherapy plus trastuzumab and pembrolizumab showed significant improvement in PFS and ORR; however, it did not prolong OS, and we are awaiting results of the final analysis of OS. Furthermore, it had negative impact on patients with CPS < 1. Notably, for patients with HER2-positive and CPS < 1, it remains to be investigated whether challenges can be addressed through combination therapy with novel anti-HER2 drugs and PD-1 inhibitors or other immune checkpoint inhibitors, such as LAG-3 inhibitors. Further clinical trials are needed to determine the effectiveness of such combination therapies. Moreover, treatment development for TBP is still an important clinical unmet need. To overcome trastuzumab resistance, appropriate patient selection, the use of novel anti-HER2 agents, and consideration of combination therapy with immune checkpoint inhibitors are warranted. On the other hand, immunotherapy is generally contraindicated for patients with autoimmune diseases; therefore, the development of treatments without immunotherapy remains important. Moreover, considering the nature of spatial and temporal heterogeneity of HER2 expression in AGC, clinical trials targeting patients with HER2-low will be significant. The clinical trials in progress are addressing these concerns, and the results are awaited.

## Figures and Tables

**Figure 1 cancers-16-01747-f001:**
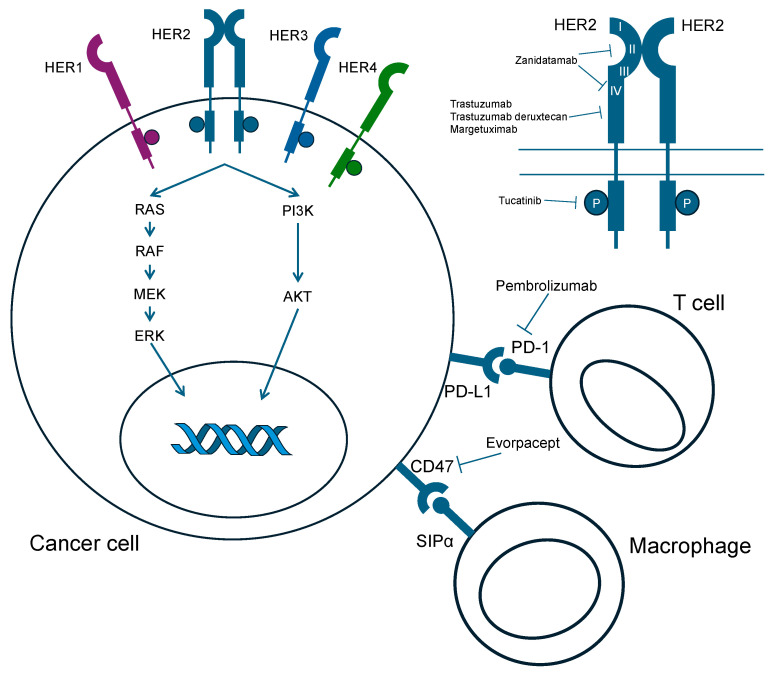
Mechanism of action of novel anti-HER2 agents.

**Table 1 cancers-16-01747-t001:** Key previous clinical trials for HER2-positive AGC.

Trial Name	Phase	Regimen	Overall Survival (months)	Progression-Free Survival (months)	Overall Response Rate (%)
First-line					
ToGA [6]	3	XP/FP + Tmab vs. XP/FP	13.8 vs. 11.1 (HR: 0.74; 95% CI: 0.60–0.91; *p* = 0.0046)	6.7 vs. 5.5 (HR: 0.71; 95% CI: 0.59–0.85; *p* = 0.0002)	47 vs. 35 (*p* = 0.0017)
TRIO-013/LoGIC [8]	3	XELOX + lapatinib vs. XELOX	12.2 vs. 11.1 (HR: 0.91; 95% CI: 0.73–1.12; *p* = 0.3492)	6.0 vs. 5.4 (HR: 0.82; 95% CI: 0.67–1.00; *p* = 0.0381)	53 vs. 39 (*p* = 0.0031)
JACOB [9]	3	CT + Tmab + pertuzumab vs. CT + Tmab	17.5 vs. 14.2 (HR: 0.84; 95% CI: 0.71–1.00; *p* = 0.057)	8.5 vs. 7.0 (HR: 0.73; 95% CI: 0.62–0.86; *p* = 0.0001)	56.7 vs. 48.3 (*p* = 0.026)
KN–811 [7]	3	CT + Tmab vs. CT + Tmab + pembrolizumab	10.0 vs. 8.1; HR: 0.72; 95% CI: 0.60–0.87; *p* = 0.0002	20.0 vs. 16.9; HR: 0.87; 95% CI: 0.72–1.06; *p* = 0.084	72.6 vs. 59.8
Previously treated with a Tmab-containing regimen					
TyTAN [10]	3	PTX + lapatinib vs. PTX	11.0 vs. 8.9 (HR: 0.84; 95% CI: 0.64–1.11; *p* = 0.1044)	5.4 vs. 4.4 (HR: 0.85; 95% CI: 0.63–1.13; *p* = 0.2441)	27 vs. 9 (*p* < 0.001)
GATSBY [11]	2/3	PTX vs. T-DM1	8.6 vs. 7.9 (HR: 1.15; 95% CI: 0.87–1.51; *p* = 0.86)	2.9 vs. 2.7 (HR: 0.84; 95% CI: 0.64–1.11; *p* = 0.1044)	19.6 vs. 20.6 (*p* = 0.8406)
T-ACT [12]	rP2	PTX vs. PTX + Tmab	10.0 vs. 10.2 (HR: 1.23; 95% CI: 0.76–1.99; *p* = 0.20)	3.2 vs. 3.7 (HR: 1.15; 95% CI: 0.87–1.51; *p* = 0.86)	32 vs. 33 (*p* = 1.00)
DESTINY-Gastric01 [13]	rP2	T-DXd vs. Physician’s choice (PTX or IRI)	12.5 vs. 8.4 (HR: 0.59; 95% CI: 0.39–0.88; *p* = 0.01)	5.6 vs. 3.5 (HR: 0.47; 95% CI: 0.31–0.71)	51 vs. 14 (*p* < 0.001)

XP: capecitabine plus cisplatin, FP: fluorouracil plus cisplatin, Tmab: trastuzumab, XELOX: capecitabine plus oxaliplatin, PTX: paclitaxel, T-DM1: trastuzumab emtansine, T-DXd: trastuzumab deruxtecan, IRI: irinotecan, rP2: randomized phase II.

**Table 2 cancers-16-01747-t002:** Ongoing clinical trials for HER2-positive AGC.

Trial Number	Phase	Experimental Arm	Control Arm	Primary Endpoint	Estimated Study Completion Date
First-line					
NCT05152147	3	XELOX/FP + Zanidatamab, XELOX/FP + Zanidatamab + Tislelizumab	XELOX/FP + Tmab	PFS, OS	1 July 2025
NCT03929666	2	Zanidatamab + FP/mFOLFOX6/XELOX	None	Part 1 Incidence of DLTs, AEs, lab abnormalities Part 2 ORR	30 October 2026
NCT05382364	1b/2	Tucatinib + Tmab + FOLFOX/XELOX, Tucatinib + Tmab + Pembrolizumab + FOLFOX/XELOX, Tucatinib + Tmab + Pembrolizumab, Tucatinib + Tmab + FOLFOX/XELOX	None	Incidence of renal DLTs, AEs, laboratory abnormalities, DLTs, and dose alterations	29 December 2025
NCT04082364	2/3	Margetuximab + Retifanlimab, Margetuximab + Retifanlimab + XELOX/mFOLFOX6, Margetuximab + Tebotelimab + XELOX/mFOLFOX6, Margetuximab + XELOX/mFOLFOX6	Tmab + XELOX/mFOLFOX6	AEs, ORR for non-MSI-H	March 2024
NCT04379596	1/2	Part 1 T-DXd + 5-FU, T-DXd + Cape, T-DXd + Durvalumab, T-DXd + XELOX, T-DXd + 5-FU + Durvalumab, T-DXd + Cape + Durvalumab. Part 2 T-DXd, T-DXd + 5-FU/Cape + Pembrolizumab, T-DXd + Pembrolizumab, T-DXd + 5-FU/Cape + Pembrolizumab. Part 3 T-DXd + 5-FU/Cape + MEDI5752	Part 1 None Part 2 Tmab + 5-FU/Cape + Cisplatin/Oxaliplatin. Part 3 None	Part 1 Occurrence of AEs and SAEs, changes from baseline in laboratory parameters, vital signs, and ECG Parts 2 and 3 ORR	30 July 2026
After Second–line					
NCT04704934	3	T-DXd	RAM + PTX	OS	15 November 2024
NCT05002127	2/3	Evorpacept + Tmab + RAM + PTX	RAM + PTX	Phase 2 part ORR Phase 3 part OS	August 2028
NCT04499924	2/3	Tucatinib + Tmab + RAM + PTX, Tucatinib + RAM + PTX	RAM + PTX	Phase 2 part Incidence of DLTs, AEs, laboratory abnormalities, dose modifications. Phase 3 part OS, PFS	31 March 2027
NCT05190445	2	Cinrebafusp alfa + RAM + PTX, Cinrebafusp + Tucatinib	None	ORR	1 February 2023
NCT04989816	2	T-DXd	None	ORR	28 February 2024
NCT05270889	2	Zanidatamab + Tislelizumab	None	ORR	June 2024

PFS: progression-free survival, OS: overall survival, ORR: overall response rate, DLT: dose-limiting toxicities, SAE: severe adverse event, XELOX: capecitabine plus oxaliplatin, FP: fluorouracil plus cisplatin, mFOLFOX6: modified fluorouracil, leucovorin, plus oxaliplatin, Non-MSI-H: nonmicrosatellite instability—high, Tmab: trastuzumab, T-DXd: trastuzumab deruxtecan, Cape: capecitabine, FU: fluorouracil, RAM: ramucirumab, PTX: paclitaxel.

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
