# Peer review of "Recent Progress in Treatment for HER2-Positive Advanced Gastric Cancer"

_cancers, 2024, doi:10.3390/cancers16091747_

Round 1

Reviewer 1 Report

Comments and Suggestions for Authors

1. What is KN-811 in Table 1? The text says "Although several clinical trials of anti-HER2 therapy, 46 which has provided survival benefit for breast cancer, were conducted in first-line setting after the ToGA trial, none showed survival benefit for HER2-positive AGC (Table 1)." but some of them look positive. Just make it clear and explain details of negative studies.

2. Is it possible to indicate the expecting date of the outcome in these on going trials in Table 2?

3. Any trials targeting HER3 other than JACOB?

Author Response

Dear Reviewer #1,

We thank the reviewers for reviewing our manuscript and appreciate the opportunity to resubmit our manuscript following revisions according to the reviewers’ comments. Please find the revised version of our manuscript enclosed. All changes in the revised manuscript have been highlighted in yellow. Please find our responses to individual comments below. We hope that our revised manuscript can be accepted for publication in Cancers.

Comment 1:

 What is KN-811 in Table 1? The text says "Although several clinical trials of anti-HER2 therapy, 46 which has provided survival benefit for breast cancer, were conducted in first-line setting after the ToGA trial, none showed survival benefit for HER2-positive AGC (Table 1)." but some of them look positive. Just make it clear and explain details of negative studies.

Response:

We appreciate your comment. We have filled in the blanks in Table 1. In addition, we have explained in more detail the presumed reasons for the negative studies.

Comment 2:

 Is it possible to indicate the expecting date of the outcome in these on going trials in Table 2?

Response:

We appreciate your comment. Does the reviewer indicate Estimated Study completion date? We added in the Table 2. We found NCT02689284 study has been already published, therefore, deleted it from Table 2.

Comment 3:

 Any trials targeting HER3 other than JACOB?

Response:

We appreciate your important comment. As the reviewer point it out, targeting HER3 agent is thought to be overcoming acquired trastuzumab resistance using Xenograft model (Cell Death Discov. 8, 478 (2022). https://doi.org/10.1038/s41420-022-01259-z, Cancer Lett. 2016 Sep 28;380(1):20-30.  doi: 10.1016/j.canlet.2016.06.005.) and a phase 2 study of using HER3-DXd monotherapy for advanced or metastatic solid tumors is ongoing (NCT06172478). However, this trial targets HER2-negative gastric of esophagogastric cancer. Therefore, we excluded this study from this manuscript. 

Reviewer 2 Report

Comments and Suggestions for Authors

Thank you for the opportunity to review the manuscript titled, "Recent Progress in Treatment for HER2-Positive Advanced Gastric Cancer". In this manuscript, the authors have reviewed the various agents targeting HER2 positivity in gastric cancer. The manuscript is well written and is suitable for publication. It would be helpful if the authors are able to incorporate a diagram depicting the sites at which different agents target the HER2 receptor.

Author Response

Dear Reviewer #2,

We thank the reviewers for reviewing our manuscript and appreciate the opportunity to resubmit our manuscript following revisions according to the reviewers’ comments. Please find the revised version of our manuscript enclosed. All changes in the revised manuscript have been highlighted in yellow. Please find our responses to individual comments below. We hope that our revised manuscript can be accepted for publication in Cancers.

Comment

Thank you for the opportunity to review the manuscript titled, "Recent Progress in Treatment for HER2-Positive Advanced Gastric Cancer". In this manuscript, the authors have reviewed the various agents targeting HER2 positivity in gastric cancer. The manuscript is well written and is suitable for publication. It would be helpful if the authors are able to incorporate a diagram depicting the sites at which different agents target the HER2 receptor.

Response:

We appreciate your comment. We also believe that a diagram showing mechanisms is helpful for readers and have submitted it as Figure 1.

Reviewer 3 Report

Comments and Suggestions for Authors

The article titled " Recent Progress in Treatment for HER2-Positive Advanced Gastric Cancer" by Takeshi Kawakami and Kentaro Yamazaki provides valuable insights. However, there are several areas that require attention:

1. Authors can explain more about that not all patients with HER2-positive advanced gastric cancer respond to combination therapy with trastuzumab and chemotherapy, and some develop resistance over time.

2. Trastuzumab deruxtecan has shown to use later-line treatment, but, this drug is greatly associated with toxicities, such as interstitial lung disease, cardiac toxicity etc. Authors must highlight these literature and debate it

3. The optimal sequencing and combination of these therapies are still uncertain, and more investigation is required to detect biomarkers that can predict response to these therapies and to develop approaches to overcome resistance. Authors must discuss it

4. Authors must include the treatment options of HER2-negative gastric cancer and needs to address effective management options

5. Authors indicate that clinical trials targeting HER2 have unsuccessful in first- or subsequent-line treatment. Thus, authors discuss the possible explanations for these failures and to find novel strategies to overcome them

6. Authors mainly focuses on the use of targeted therapies and immunotherapy. But, authors can consider to explain the other treatment methods, such as radiation therapy, surgery, and palliative care

7. This combination therapy may not be appropriate for all patients, particularly those with pre-existing autoimmune disorders or other contraindications to immunotherapy. Authors can comprise it.

8. Authors can discuss the possible effect of these combination treatment options on patient quality of life, treatment cost, access to care, and patient preferences and possible benefits and drawback of these therapies.

Comments on the Quality of English Language

Moderate editing of English language required

Author Response

Dear Reviewer #3,

We thank the reviewers for reviewing our manuscript and appreciate the opportunity to resubmit our manuscript following revisions according to the reviewers’ comments. Please find the revised version of our manuscript enclosed. All changes in the revised manuscript have been highlighted in yellow. Please find our responses to individual comments below. We hope that our revised manuscript can be accepted for publication in Cancers.

Comment 1:

 Authors can explain more about that not all patients with HER2-positive advanced gastric cancer respond to combination therapy with trastuzumab and chemotherapy, and some develop resistance over time.

Response:

We appreciate your comment. As the reviewer pointed out, several patients have initial and acquired resistance for trastuzumab because of heterogenous HER2 expression, and/or emergence of MET amplification, or PIK3CA mutation. We described resistance mechanism and clinical trials expected to overcome resistance in “Previously treated with a trastuzumab-containing regimen after second-line treatment”. The timing of resistance is unknown, therefore, we added in “Trastuzumab First-line” part.

Comment2:

 Trastuzumab deruxtecan has shown to use later-line treatment, but, this drug is greatly associated with toxicities, such as interstitial lung disease, cardiac toxicity etc. Authors must highlight these literature and debate it

Response:

We appreciate your important comment. As the reviewer point out, cardiac toxicity is one of the important adverse events of trastuzumab, however, the proportion of cardio toxicity of trastuzumab deruxtecan is relatively low. ILD and nausea/vomiting are considered important adverse events in the management of trastuzumab deruxtecan. We discuss these adverse events in more detail (highlighted yellow color).

Comment 3:

 The optimal sequencing and combination of these therapies are still uncertain, and more investigation is required to detect biomarkers that can predict response to these therapies and to develop approaches to overcome resistance. Authors must discuss it.

Response:

We appreciate your comment. The optimal sequencing and combination remains an important clinical question. Chemotherapy plus trastuzumab with pembrolizumab is considered as first-line treatment and trastuzumab deruxtecan is considered as third-line treatment. When the results of DESTINY-Gastric04 study comparing trastuzumab deruxtecan with paclitaxel plus ramucirumab as second-line treatment are available, we will need to discuss when is the best time to administer trastuzumab deruxtecan. Unfortunately, few biomarkers have been implicated in resistance mechanisms and are in clinical trials other than those mentioned in the maintext. Clinical trials of Evorpacept targeting innate immunity are expected to overcome resistance and are described in “Previously treated with a trastuzumab-containing regimen after second-line treatment” section.

Comment 4:

 Authors must include the treatment options of HER2-negative gastric cancer and needs to address effective management options.

Response:

We appreciate your valuable comment. The topic of this manuscript is targeting HER2-positive AGC, therefore, we did not mention HER-negative AGC to avoid readers misunderstanding the content of this review. However, it is important information that chemotherapy plus immune checkpoint inhibitor is effective for HER2-negative AGC, therefore, we added it in the main text (highlighted yellow color, line ○○).

Comment 5:

 Authors indicate that clinical trials targeting HER2 have unsuccessful in first- or subsequent-line treatment. Thus, authors discuss the possible explanations for these failures and to find novel strategies to overcome them.

Response:

We appreciate your important comment. We discuss the possible explanations about the resistant mechanisms and novel strategies to overcome them in line 106 to 111 and in “Previously treated with a trastuzumab-containing regimen after second-line treatment” section.

Comment 6:

 Authors mainly focuses on the use of targeted therapies and immunotherapy. But, authors can consider to explain the other treatment methods, such as radiation therapy, surgery, and palliative care

Response:

We appreciate your comment. The theme of this review article is the treatment development for HER2-positive AGC. This review discusses the development of anti-HER2 therapies for patients who are eligible for systemic chemotherapy. Therefore, surgery, radiotherapy, and palliative treatment are not discussed in this review.

Comment 7:

 This combination therapy may not be appropriate for all patients, particularly those with pre-existing autoimmune disorders or other contraindications to immunotherapy. Authors can comprise it.

Response:

We appreciate your valuable comment. As the reviewer points out, patients with autoimmune disease have difficulty receiving combination therapy with immune checkpoint inhibitors. We added this to the conclusion and future direction part as an issue for treatment development.

Comment 8:

 Authors can discuss the possible effect of these combination treatment options on patient quality of life, treatment cost, access to care, and patient preferences and possible benefits and drawback of these therapies.

Response:

We appreciate your important comment. We have added the information on published quality of life data (highlighted yellow color in “Trastuzumab First-line” part). Treatment costs and access to care vary in each country because of different insurance systems, and it is very difficult to describe the situation in the main text. Patient preferences, benefits and drawbacks of treatment for HER2-positive AGC are considered as efficacy and risk of adverse event occurrence and are discussed in the text.

Reviewer 4 Report

Comments and Suggestions for Authors

The individualized selection of targeted drugs for stomach cancer remains one of the most significant approaches in drug therapy of cancer. It is worth noting that HER2 therapy is indicated for tumors of this localization, but its effectiveness still remains low, due to the lack of standard approaches, different sensitivity to drugs and other unexplained reasons.

Comments on the Quality of English Language

The individualized selection of targeted drugs for stomach cancer remains one of the most significant approaches in drug therapy of cancer. It is worth noting that HER2 therapy is indicated for tumors of this localization, but its effectiveness still remains low, due to the lack of standard approaches, different sensitivity to drugs and other unexplained reasons.

Author Response

Dear Reviewer #4,

We thank the reviewers for reviewing our manuscript and appreciate the opportunity to resubmit our manuscript following revisions according to the reviewers’ comments. Please find the revised version of our manuscript enclosed. All changes in the revised manuscript have been highlighted in yellow. Please find our responses to individual comments below. We hope that our revised manuscript can be accepted for publication in Cancers.

Comment:

The individualized selection of targeted drugs for stomach cancer remains one of the most significant approaches in drug therapy of cancer. It is worth noting that HER2 therapy is indicated for tumors of this localization, but its effectiveness still remains low, due to the lack of standard approaches, different sensitivity to drugs and other unexplained reasons.

Response:

We appreciate your insightful comment. I totally agree with your idea.

Round 2

Reviewer 3 Report

Comments and Suggestions for Authors

 Accept in present form

Comments on the Quality of English Language

Moderate editing of English language required